# Impact of *Lactobacillus*- and *Bifidobacterium*-Based Direct-Fed Microbials on the Performance, Intestinal Morphology, and Fecal Bacterial Populations of Nursery Pigs

**DOI:** 10.3390/microorganisms12091786

**Published:** 2024-08-28

**Authors:** Juan Castillo Zuniga, Anlly M. Fresno Rueda, Ryan S. Samuel, Benoit St-Pierre, Crystal L. Levesque

**Affiliations:** Animal Science Complex, Department of Animal Science, South Dakota State University, P.O. Box 2170, Brookings, SD 57007, USA; juan.castillo@sdstate.edu (J.C.Z.); anlly.rueda@sdstate.edu (A.M.F.R.); ryan.samuel@sdstate.edu (R.S.S.); benoit.st-pierre@sdstate.edu (B.S.-P.)

**Keywords:** weaning, swine, intestinal morphology, microbiota, bacteria

## Abstract

Weaning is a critical stage in the swine production cycle, as young pigs need to adjust to sudden and dramatic changes in their diet and environment. Among the various organ systems affected, the gastrointestinal tract is one of the more severely impacted during this transition. Traditionally, challenges at weaning have been managed by prophylactic use of antibiotics, which not only provides protection against diarrhea and other gut dysfunction but also has growth-promoting effects. With banning or major restrictions on the use of antibiotics for this purpose, various alternative products have been developed as potential replacements, including direct-fed microbials (DFMs) such as probiotics and postbiotics. As their efficiency needs to be improved, a continued effort to gain a deeper understanding of their mechanism of action is necessary. In this context, this report presents a study on the impact of a *Lactobacillus*-based probiotic (LPr) and a *Bifidobacterium*-based postbiotic (BPo) when added to the diet during the nursery phase. For animal performance, an effect was observed in the early stages (Day 0 to Day 10), as pigs fed diets supplemented with either DFMs were found to have higher average daily feed intake (ADFI) compared to pigs fed the control diet (*p* < 0.05). Histological analysis of intestinal morphology on D10 revealed that the ileum of supplemented pigs had a higher villus height/crypt depth ratio (*p* < 0.05) compared to controls, indicating a benefit of the DFMs for gut health. In an effort to further explore potential mechanisms of action, the effects of the DFMs on gut microbial composition were investigated using fecal microbial communities as a non-invasive representative approach. At the bacterial family level, Lactobacillaceae were found in higher abundance in pigs fed either LPr (D10; *p* < 0.05) or BPo (D47; *p* < 0.05). At the Operational Taxonomic Unit (OTU) level, which can be used as a proxy to assess species composition, Ssd-00950 and Ssd-01187 were found in higher abundance in DFM-supplemented pigs on D47 (*p* < 0.05). Using nucleotide sequence identity, these OTUs were predicted to be putative strains of *Congobacterium massiliense* and *Absicoccus porci*, respectively. In contrast, OTU Ssd-00039, which was predicted to be a strain of *Streptococcus alactolyticus*, was in lower abundance in BPo-supplemented pigs on D47 (*p* < 0.05). Together, these results indicate that the DFMs tested in this study can impact various aspects of gut function.

## 1. Introduction

In intensive swine production systems, optimizing gut function is critical for ensuring both nutrition and animal health. This is particularly true during weaning, as it represents a critical stage of adaptation to major and abrupt changes that young pigs experience [1]. Indeed, from the switch in diet to dealing with multiple stressors such as separation from the dam, settling into an unfamiliar environment, and adjustment to a new social hierarchy, weaning can have short- and long-term consequences on gut function, which ultimately can impact productivity and profitability [2,3].

The change in diet combined with reduced feed intake at weaning impairs the expression of digestive enzymes as well as the absorptive capacity of the epithelial lining [2,4,5,6,7,8,9]. In addition, the gut of weaned pigs is affected by increased permeability, alterations in enteric neuron numbers and cholinergic activity, as well as by upregulation of proinflammatory cytokines and hyperplasia of intestinal mast cells [10]. Together, these factors contribute to increased disease susceptibility in young pigs. For instance, impaired host digestion and absorption provide a source of nutrients for opportunistic pathogens, whose colonization and proliferation are facilitated by a compromised immune response. These conditions can lead to further damage from impaired gut barrier function, triggering the onset of diarrhea or other digestive dysfunctions [11,12,13], which ultimately result in economic losses to producers [5,6,14].

Traditionally, diarrhea and digestive dysfunctions when weaning have been managed through the prophylactic use of antibiotics. Under these conditions, antibiotics also exhibit growth-promoting effects. However, higher incidences of antibiotic resistance in food animals, as well as increased consumer concerns, have led to the implementation of policies that restrict or ban the use of antibiotics as growth promoters [15]. While the mechanisms responsible for the growth-promoting effects of antibiotics remain poorly understood [16], they are likely to involve modulation of the gut microbiota, since these agents act as antimicrobials. This is consistent with a growing body of literature reporting that the composition of gut microbial communities can be associated with the health status and performance of individual animals [13,17,18]. Thus, compounds or agents that can change gut symbiont profiles have the potential to be developed as tools to improve critical livestock production parameters.

To date, a number of candidate alternatives to antibiotics have been developed. These include formulations based on plant compounds and extracts, such as prebiotics, essential oils, or other phytochemicals, as well as microbial-based products, often referred to as direct-fed microbials (DFMs), which include probiotics and postbiotics. Supplementation of nursery diets with probiotics, which consist of live microorganisms [19], is thought to reduce pathogen proliferation by establishing beneficial bacteria and lowering gut pH, as well as to enhance gut health and barrier function by providing substrates such as short-chain fatty acids (SCFAs) to host epithelial cells [20]. Postbiotics, which consist of microbial products derived from probiotics grown in culture [21], contain active compounds with antimicrobial, immunomodulatory, and/or anti-inflammatory properties. Postbiotics are considered preferable to probiotics under certain conditions, as they contain the same beneficial components, in the form of lactic acid, bacteriocins, short-chain fatty acids, exopolysaccharides, and cell wall fragments, without concerns such as having to ensure the survival of live organisms or the potential production of undesirable compounds when exposed to the gut environment [21].

In this context, the study presented in this report aimed to evaluate the effects of feeding a *Lactobacillus*-based probiotic or a *Bifidobacterium*-based postbiotic to weaned pigs until the beginning of the grow-finisher phase. Overall, the DFMs tested were beneficial in terms of performance and intestinal morphology, and they were found to impact the fecal bacterial composition of pigs.

## 2. Materials and Methods

### 2.1. Animals, Housing, and Diets

The trial described in this report was conducted at the South Dakota State University (SDSU) Off-Site Wean-to-Finish Barn, with all procedures approved by the SDSU Institutional Animal Care and Use Committee (IACUC) before the start of the study. A total of 1040 pigs (initial body weight (BW) 6.1 ± 0.1 kg) were selected based on overall health by visual assessment from an initial population of 1244 weaned pigs (PIC 800 × PIC) that were delivered to the facility from a single local producer. Selected pigs were allocated to 40 pens, blocked by weight and barn location, in order to populate individual pens with 26 pigs (13 barrows and 13 gilts per pen). Each pen had an area of 3.1 m × 6.9 m, with approximately 0.82 m^2^ for each pig. The experimental design was a complete block, with 10 blocks per treatment and every treatment represented in every section of the barn. Diets were formulated to provide lysine and energy that met or exceeded the NRC (2012) recommendations for each diet ([22]; Appendix A). Pens were assigned to one of four dietary treatments: Control diet (Control), Control diet + 0.1% of a *Lactobacillus*-based probiotic (LPr-0.1%), Control diet + 0.2% of a *Lactobacillus*-based probiotic (LPr-02%), or Control diet + 0.2% of a *Bifidobacterium*-based postbiotic (BPo-0.2%). They were each included in their respective treatment diets starting at the very beginning of the nursery phase (D0, Phase 1) until D47, which corresponded to the beginning of the grower phase (Phase 3). LPr and BPo were both manufactured according to proprietary protocols. The LPr product consisted of two proprietary *Lactobacilli* strains at 2 × 10^9^ CFU/g.

### 2.2. Growth Performance Measurements and Calculations

Pigs were weighed by pen at barn entry (D0), as well as on D10 (end of Phase 1), D21 (Phase 2), D47 (beginning of Phase 3, last day of DFM supplementation), D70, D105, and D135. Each pen used in the trial was equipped with one dry feeder and two cup waterers for ad libitum access to feed and water, respectively. A Feedlogic M-Series system (Feedlogic ComDel Innovation, Wilmar, MN, USA) was used for dispensing feed. On each weigh day, the amount of feed remaining was determined using an in-house calibration curve by measuring the distance from the top of the feeder to the top of the leveled feed; the calibration curve was designed to also correct for feed density. Feed disappearance was then calculated by subtracting the remaining feed from the total feed delivered since the previous weigh day, which had been recorded by the Feedlogic software (v.2.4.3).

Average daily gain (ADG), average daily feed intake (ADFI), and gain-to-feed ratio (G/F) were calculated using the following:$$ADG=\frac{\left(current\;pen\;weight-previous\;pen\;weight+removed\;pig\;weights\right)}{pig\;days}$$
$$ADFI=\frac{\left(feed\;delivery-remaining\;feed\right)}{pig\;days}$$
$$G/F=\frac{ADG}{ADFI}$$

To avoid compromising the calculations of performance results, ‘pig days’ took into account the removal date(s) and weight(s) of any pig(s) between weigh days.

### 2.3. Histological Analysis

Samples from a total of 40 pigs (*n* = 10/treatment) harvested on D10 were analyzed to determine the potential effect of DFMs on intestinal morphology; one representative from each pen was randomly selected based on weight and health status. Selected pigs were euthanized using a non-penetrating captive bolt gun (CASH^®^, Frontmatec Accles & Shelvoke Ltd., Sutton Coldfield, UK), which was applied to deliver a concussive force sufficient to cause effective stunning; this procedure was performed according to guidelines for the humane euthanasia of swine as prescribed by the American Association of Swine Veterinarians (AASV) and the National Pork Board (NPB). Euthanasia was confirmed by verifying that respiration had ceased and that no corneal reflex was present before proceeding to sample collection. The intestinal tract was dissected and removed from the body cavity and then samples 5 cm in length were collected from the jejunum (midpoint of the small intestine) and ileum (10 cm proximal from the ileocecal junction).

Intestinal samples were fixed, sectioned, and then stained with hematoxylin and eosin at the SDSU Animal Disease Research and Diagnostic Laboratory. Villus height (VH), measured from the top of the villus to the villus–crypt junction, and crypt depth (CD), measured from the villus–crypt junction to the base, were determined at 4× magnification using a Micromaster^®^ microscope (Fisher Scientific, Waltham, MA, USA) equipped with a 0.55× wifi camera eyepiece (MC500-W 3rd Gen., Meiji Techno Co., Ltd., Saitama, Japan) and Micro-Capture software (v.6.9.11) (Meiji Techno Co., Ltd., Saitama, Japan). The average of ten measurements from each sample was used as a representative of each pen.

### 2.4. Isolation of Microbial Genomic DNA and Sequencing of 16S rRNA Gene Amplicons

On D10 and D47, fecal samples were collected by rectal palpation from one representative pig in each pen for the Control, LPr-02%, and BPo-02% groups. The same pigs were sampled at each time point. Samples (total *n* = 60) were stored frozen at −20 °C until further processed. Microbial genomic DNA was extracted from individual samples using a bead-beating plus column approach [23] using the QIAamp DNA Mini Kit (Qiagen, Hilden, Germany). The V1–V3 regions of the bacterial 16S rRNA gene were selected for analysis and targeted by PCR using the universal forward 27F 5′AGAGTTTGATCMTGCTCAG and reverse 519R 5′GWATTACCGCGCGCGCTG primers [24]. Purified microbial genomic DNA samples were processed by Molecular Research DNA (MRDNA, Shallowater, TX, USA) for V1–V3 amplification, and for amplicon sequencing with the Illumina MiSeq 2 × 300 platform to generate overlapping paired-end reads.

### 2.5. Bioinformatics Analyses

A combination of custom-written Perl scripts and publicly available software was employed for 16S rRNA data analysis. The overlapping paired-end reads generated from the same V1–V3 amplicons were first merged, then screened for quality by filtering for (1) the presence of both intact 27F and 519R primer sequences, (2) a minimal average Phred quality score of Q33, and (3) a length between 400 and 580 nt [25]. Quality-filtered sequences were then aligned and clustered into Operational Taxonomic Units (OTUs) using a threshold of 4% sequence dissimilarity. This cutoff is more suitable for the V1–V3 hypervariable regions than the typical 3% threshold that is commonly used for 16S rRNA sequence data clustering [26,27]. The OTUs were then screened using three independent approaches to flag artifacts needing to be removed [25]. Identification of chimeric sequences was performed first, with the commands ‘chimera.slayer’ [28] and ‘chimera.uchime’ [29] using the open-source software package MOTHUR (v.1.44.1) [30]. A sequence-alignment-based approach with BLAST [31] was then used on the remaining OTUs to assess the quality of their ends; OTUs were considered artifacts when more than five nucleotides were missing from the 5′ or 3′ ends of their respective alignments with their closest match of equal or longer sequence length from the NCBI ‘nt’ database. For OTUs with only one or two assigned reads, an additional screen for artifacts was performed, where only OTUs showing a perfect or near-perfect match to a sequence in the NCBI ‘nt’ database were kept for analysis; a maximum of 1% dissimilarity was tolerated. All OTUs and their assigned reads that were flagged during these screens were removed from further analyses. For all artifact-free OTUs, phylum and family-level affiliations were determined using the RDP Classifier [32]. For the most abundant OTUs, the closest valid relatives were also identified using BLAST against the ‘refseq_rna’ database [31].

The alpha diversity indices ‘Observed OTUs’, ‘Chao’, ‘Ace’, and ‘Shannon’ were generated using the ‘summary.single’ command in MOTHUR (v.1.44.1) [30]. For beta diversity analysis, Bray–Curtis distances were calculated first, then used as input for principal coordinate analysis (PCoA), using the commands ‘summary.shared’ then ‘pcoa’, respectively, in MOTHUR (v.1.44.1) [30]. A PCoA plot was created using the Tableau Visualization Software (Version 2020.4, https://www.tableau.com/products/new-features accessed on 22 April 2024).

### 2.6. Statistical Analyses

Data for BW, ADG, ADFI, G/F, and histological measurements were analyzed using the PROC GLIMMIX procedure of SAS (Version 9.4, SAS Inst. Inc., Cary, NC, USA), with ‘dietary treatment’ as the main effect and ‘pen’ as the experimental unit. Statistical testing of microbiome data was performed in ‘R’ (Version 3.6.0). Analysis of variance (ANOVA) with a post hoc ‘HSD.test’ pairwise function was used to compare alpha diversity indices. The Kruskal–Wallis rank-sum test was performed for the analysis of taxonomic groups and most abundant OTUs (non-parametric data), concomitant with the pairwise Wilcoxon rank-sum test, which included the Benjamini–Hochberg correction for controlling the false discovery rate. For all statistical tests described above, a threshold of *p* ≤ 0.05 was considered significant.

## 3. Results

### 3.1. Animal Performance and Intestinal Morphology

The potential effects of the DFMs on animal performance were assessed by comparing the BW, ADG, ADFI, and G/F of the supplemented groups with the control group (Table 1, Table 2, Table 3 and Table 4). Of these metrics, only ADFI was found to vary, with higher values in all three DFM-supplemented groups compared to the controls during the D0–D10 period (*p* < 0.05).

Gut integrity was assessed by measuring intestinal VH and CD. Histological analysis of ileal tissue on D10 revealed that pigs fed a diet supplemented with either of the DFMs had a greater VH/CD ratio compared to pigs fed the control diet (*p* < 0.05; Table 5). For jejunal tissue, while the samples from DFM-fed pigs showed longer VH, shorter CD, and higher VH/CD ratios compared to controls, these differences were not significant (*p* > 0.05; Table 5).

### 3.2. Bacterial Composition Analysis

To determine the impact of DFM supplementation on gut microbial communities in pigs, an analysis of bacterial composition was performed on D10 and D47 pigs from the Control, LPr-02%, and BPo-02% groups using fecal samples as a proxy. After quality filtering, a combined total of 1,552,212 reads were analyzed, with the mean sequence yield amongst groups ranging between 20,465.1 and 32,692.3 high-quality reads per sample (Appendix A).

#### 3.2.1. Taxonomic Composition

At the phylum level, Bacillota (formerly known as Firmicutes) were by far the most highly represented, with an average abundance of 90.54% across all samples (Table 6). Six families (Lachnospiraceae, Lactobacillaceae, Streptococcaceae, Ruminococcaceae, Clostridiaceae 1, Erysipelotrichaceae, and Clostridiales Incertae Sedis XIII) were identified as most highly represented amongst Bacillota. Bacteroidota (formerly known as Bacteroidetes) were found to be the second most abundant phylum in this study, with an average representation of 4.65% across all samples. Notably, Prevotellaceae was the most prominent Bacteroidota family, with the highest levels observed on D47, where they represented at least 91.85% of Bacteroidota sequences across all samples at that time point. Actinomycetota (formerly known as Actinobacteria), with an average abundance of 1.22% across all samples, as well as nine other minor phyla for which the combined average representation per sample was less than 1% (0.57%), were the remaining sequences that could be assigned to a recognized phylogenetic lineage in this study.

Further analyses of the taxonomy data revealed two distinct composition patterns. The first indicated that the most dramatic differences were between D10 and D47 samples. For instance, higher abundances of Lactobacillaceae, Streptococcaceae, and Prevotellaceae were observed in D47 samples compared to D10 samples, while Lachnospiraceae and Clostridiales Incertae Sedis XIII showed the opposite composition pattern (*p* < 0.05). Notably, the respective group abundances of Streptococcaceae were between 47.9× and 206.9× higher in D47 samples than in D10 samples when comparing pairs of samples from the same dietary treatment. The other composition pattern observed consisted of differences between a group fed a diet supplemented with a DFM and its corresponding control group from the same time point. In this category, Lactobacillaceae were found in higher abundance in the BPo group on D10 and in the LPr group on D47, while Clostridiales Incertae Sedis XIII were in higher abundance in the LPr group on D47 (*p* < 0.05). In contrast, Streptococcaceae were in significantly lower abundance in the BPo group on D47 (*p* < 0.05).

#### 3.2.2. Alpha and Beta Diversity

In light of the differences in taxonomic composition observed, further analyses were pursued using OTUs as a proxy to investigate bacterial species composition. A total of 9021 OTUs were identified across all samples (Appendix A). While numerical differences were observed for the alpha diversity indices analyzed, no statistical differences were detected (*p* > 0.05; Table 7). To gain further insights, data were also investigated for beta diversity using principal coordinate analysis (PCoA) using a Bray–Curtis distance-matrix-based methodology. The results from this approach (Figure 1) were consistent with the taxonomy analysis described above, as the only significant differences found were associated with the timing of sample collection.

#### 3.2.3. OTU Composition

While taxonomy and beta diversity analyses indicated a very strong association between sample collection time point and fecal bacterial composition, the potential effects of the DFMs under study on the swine gut microbiome appeared to have been more subtle. Thus, further analyses of the most abundant bacterial OTUs, defined as being present at an average of at least 1% in at least one group, were performed as a strategy to gain more insights. Notably, all 34 most highly represented OTUs (Table 8 and Table 9) had previously been identified in other reports [25,33,34,35,36], with the means of their combined abundances within each group ranging between 63.21% and 72.58% in this study. At least ten of these most abundant OTUs were predicted to correspond to uncharacterized bacterial species, as their respective sequence identities to the 16S rRNA genes of their corresponding closest valid relatives were less than 97% (Appendix A). While the representation of 16 of the most abundant OTUs was not different across groups (*p* > 0.05; Table 9), the remaining 18 OTUs were found to vary (*p* < 0.05; Table 8). As expected from the aforementioned taxonomy and beta diversity analyses, most differences were observed between the D10 and D47 samples. More specifically, OTUs Ssd-00001, Ssd-00003, Ssd-00011, Ssd-00014, Ssd-00039, and Ssd-00134 were found in higher abundance in D47 samples compared to D10 samples (*p* < 0.05), while OTUs Ssd-00064, Ssd-00308, and Ssd-00577 showed the opposite composition pattern (*p* < 0.05). Differences in representation between at least one DFM-supplemented group and the control group were observed on D47 for three OTUs: Ssd-00039, Ssd-00950, and Ssd-01187. Notably, the mean abundance of the latter was found to be at least 10 times higher in LPr-D47 or BPo-D47 than in CO-D47 groups (*p* < 0.05), while no differences in this OTU were observed between the control groups at different time points (CO-D10 vs. CO-D47: *p* > 0.05). Based on their level of nucleotide sequence identity, Ssd-00039, Ssd-00950, and Ssd-01187 were predicted to be strains of *Streptococcus alactolyticus*, *Congobacterium massiliense*, and *Absicoccus porci*, respectively (Appendix A).

## 4. Discussion

In recent years, an increasing number of commercial feed additives have been developed with the aim of optimizing gut function, which has long been recognized as critical in ensuring animal health and performance [1,2,37]. Feed additives encompass a wide range of products, from formulations based on plant compounds and extracts such as prebiotics, essential oils, or other phytochemicals, to microbial-based products, often referred to as direct-fed microbials, which include probiotics and postbiotics. In the latter category, a wide spectrum of bacterial strains have been developed for use as DFMs, with many affiliated with species in the genera *Lactobacillus*, *Bacillus*, *Enterococcus*, *Bifidobacterium*, or *Clostridium* [38]. In this study, the effects of a *Lactobacillus*-based probiotic and a *Bifidobacterium*-based postbiotic were tested in pigs through dietary supplementation, which started right after weaning and lasted for the duration of the nursery phase. Overall, the DFMs tested provided benefits in terms of performance and intestinal morphology, and they were found to impact the fecal bacterial composition of pigs.

To date, reported effects of DFMs on growth performance have been varied and inconsistent overall. Indeed, the inclusion of probiotics has been found to either decrease, increase, or have no significant effect on ADG, feed conversion ratio, or ADFI in weaned pigs. For instance, it was reported that a *Lactobacillus reuterii*-based probiotic resulted in higher ADG with no significant effect on ADFI [39], which is in contrast to the results described in this report; perhaps differences in experimental design could explain these contrasting outcomes, such as fewer pigs per pen and fewer pigs enrolled in the study by Yi et al. (2018), as well as comparison of the probiotic treatment group to a control group whose diet was supplemented with two antibiotics [39]. Intriguingly, supplementation with a probiotic formulation consisting of two different strains, one affiliated with *Lactobacillus* spp. and the other with *Bifidobacterium thermacidophilum*, resulted in multiple performance metrics (BW, ADG, and ADFI) being significantly higher [40]; notably, the strains used in that study had originally been isolated from fresh feces collected from Tibetan pigs, and would therefore correspond to normal residents in the swine gut [40]. However, strains of bacterial species from other microbial habitats, such as soils, have also been reported to be beneficial when used as DFMs. For instance, pigs supplemented with a *Bacillus toyonensis*-based product had higher BWs and a lower feed conversion ratio compared to the control group [41]. A formulation combining strains of *Bacillus subtilis* and *Bacillus amyloliquefaciens* was also reported to be effective, as pigs fed the DFM had greater ADG and G/F than pigs fed the control diet [42].

While varying effects on performance could be explained by the wide range of bacterial species that are used to generate DFM strains, the same level of variation in effects can be observed when comparing microbial products developed from the same bacterial species. For instance, strains of *Clostridium butyricum*, a butyrate-producing member of the swine gut microbiome [43], have been tested as DFMs in a number of independent studies, with mixed and varying impacts on performance ranging from a lack of observable effects [44] to higher ADG [45]. One possible variable could be the health status of the pigs on trial. Indeed, feeding a *C. butyricum*-based DFM to lipopolysaccharide (LPS)-challenged pigs resulted in higher final BWs [46], and a number of DFMs have been reported as beneficial in mitigating the impacts of rotavirus infections [47]. At this point, it also remains unclear to what extent factors such as DFM inclusion levels, properties of the bacterial strains, and DFM formulation can affect performance. Additional research is clearly needed to elucidate the varying outcomes that DFM supplementation can have on pig performance.

Low feed intake is one of the main causes of reduced performance during early weaning [48]. Indeed, as they adapt to a solid diet that is less digestible and palatable, a large proportion of pigs have severely reduced metabolizable energy intake during this period [49], and it may take up to two weeks for them to recover to pre-weaning energy intake levels. In addition to a direct reduction in nutrient availability, low feed intake also causes the shortening of villi, hyperplasia of crypt cells, as well as damage to gut mucosal integrity [5,50,51]. Since these morphological changes negatively impact the absorption capacity of the gut, they further impair the nutrition and performance of weaned pigs. 

In contrast to their varying effects on performance, DFMs have been more consistently associated with improved intestinal morphology, expressed in the form of longer villi, reduced crypt depth, as well as higher VH/CD ratios. These beneficial effects on intestinal tissue have been reported for a wide range of DFMs, including yeast-based products [52], *C. butyricum* [53,54,55,56], *Enterococcus faecalis* [57], *Lactiplantibacillus plantarum* [58,59], *L. reuteri* [39], *L. salivarius* [60], *B. subtilis* [61], and *B. amyloliquefaciens* [62]. Product formulations containing a combination of DFM strains such as *B. subtilis* and *B. amyloliquefaciens* [42], *C. butyricum* and *E. faecalis* [63], as well as others consisting of more complex mixtures [64,65], have also proven effective. These reported benefits to intestinal morphology align well with the histological analysis of pigs that were fed either a *Lactobacillus*-based probiotic or a *Bifidobacterium*-based postbiotic in this study.

Conceptually, DFMs would be expected to safely pass through the acid environment of the stomach and then act by changing or modulating the composition and/or activity of the gut microbiota. Considering the chemical and biological intricacies of the gut microbial environment as well as the broad spectrum of strains and species that have been developed as DFMs, the range of potential mechanisms that could be involved is likely very complex. Accordingly, to date, available reports on the impact of DFMs on gut microbial composition indicate that a range of different responses can be expected. Among possible outcomes, *Lactobacillus*-related or affiliated symbionts so far represent one of the main types of gut bacterial groups that can be modulated by DFMs [46,55,56,64,66], an effect that was also observed in this study. This finding was not surprising, considering the importance of these lactic acid-producing bacteria for gut health. Indeed, several effects that are beneficial to the host have been reported for *Lactobacilli*, such as preventing adhesion of pathogens to the gut mucosa, suppressing the growth of intestinal pathogens, and/or promoting the colonization of beneficial bacteria [67,68,69,70,71,72,73]. *Lactobacilli* have also been reported to enhance immune functions [74,75].

In addition to *Lactobacilli*, bacterial groups from other lineages have also been reported to be impacted by DFMs, including *Bifidobacterium* [54] and Erysipelotrichaceae [53], as well as more complex combinations [40,44,76,77]. The identification, in this study, of three OTUs that were not affiliated with *Lactobacilli* and whose abundances differed in treated pigs is thus consistent with these reported findings. Based on the V1–V3 region of their 16S rRNA genes, Ssd-00950 and Ssd-01187 were predicted to be strains of *C. massiliense* [78] and *A. porci* [79], respectively, which were both originally isolated from gut environments. At the time of submission of this report, *A. porci* has been reported as a Gram-positive obligate anaerobe that can ferment glucose into lactate, acetate, and formate [79]; while information on cellular or metabolic characteristics of *C. massiliense* was not available. As both OTUs were found in significantly higher abundance in pigs fed a DFM, it is possible that compounds released from the feed additives provided a growth or survival advantage. However, future research will be necessary to determine the mechanism of action involved. Ssd-00039, the third OTU showing a difference in abundance in DFM-supplemented pigs, was predicted to be a strain of *S. alactolyticus*, a predominant commensal in the swine hindgut [80,81,82] that can produce lactate from fermenting cellobiose, maltose, galactose, fructose, glucose, or mannose [80,81]. The high representation of Ssd-00039 in D47 samples could then be attributed to the likely abundance of these compounds in feed ingredients, which is supported by a previous report that this OTU was present in higher abundance in nursery pigs fed a mannose oligosaccharides-based feed-additive [25]. However, as currently available information is not sufficient to explain how DFM supplementation can result in a lower representation of Ssd-00039, additional investigations will be necessary to gain further insights.

## 5. Conclusions

The results presented in this report reveal that supplementation with individual DFMs can benefit performance and gut health in nursery pigs. While no effect of the DFMs was observed for BW, ADG, or G/F, ADFI was found to be higher in DFM-supplemented pigs in the first ten days of the nursery phase. Even if limited in duration, higher feed intake in combination with higher ileal VH/CD could translate into an economic return that would warrant the use of these DFMs under certain conditions. DFM supplementation could for instance benefit swine operations that experience health or nutritional challenges in the early stages of the nursery phase.

While recognizing that the effect of sampling time greatly outweighed the effects of DFM supplementation on the composition of fecal microbial communities, the investigations presented in this report revealed that the respective abundances of three bacterial OTUs were different in DFM-fed pigs compared to controls. Acknowledging that future investigations will be necessary to determine the roles of these candidate strains in the gut environment, the context of their identification nonetheless indicates that they may contribute to the observed effects of the DFMs. As all three OTUs have previously been identified by our group [25,33,34,36], these data would together suggest that they are common residents in the swine gut that may provide benefits for their host. In this context, the results of this study then also contribute to further elucidation of gut microbial communities and how specific members can contribute to the health and performance of their hosts [13,17,18].

## Figures and Tables

**Figure 1 microorganisms-12-01786-f001:**
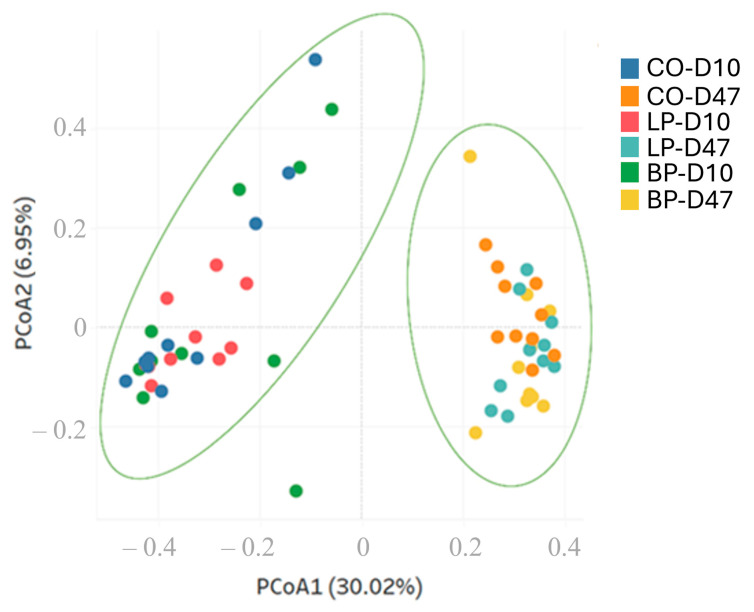
Comparison of fecal bacterial communities from pigs under three different dietary treatments using principal coordinate analysis (PCoA). PCoA was performed using a Bray–Curtis distance matrix. The *x* and *y* axes correspond to principal components 1 (PCoA1) and 2 (PCoA2). Green ellipses represent differences between D10 and D47 samples.

**Table 1 microorganisms-12-01786-t001:** Mean body weights (BWs, kg) of pigs from weaning to market weight in four dietary treatment groups: Control diet, Control diet + 0.1% of a *Lactobacillus*-based probiotic, Control diet + 0.2% of a *Lactobacillus*-based probiotic, and Control diet + 0.2% of a *Bifidobacterium*-based postbiotic.

Time Point	Control	LPr-0.1% ^1^	LPr-0.2% ^2^	BPo-0.2% ^3^	SEM	*p*-Value
D0	6.0	6.1	6.2	6.1	0.1	0.64
D10	7.5	7.6	7.6	7.5	0.1	0.95
D21	11.3	11.2	11.4	11.3	0.2	0.93
D47	27.3	27.1	27.3	27.4	0.4	0.96
D70	48.5	48.8	48.7	49.1	0.6	0.92
D105	84.3	84.1	84.3	84.8	1.0	0.96
D135	114.5	114.6	114.1	114.2	1.3	0.99

^1^ LPr-0.1% = Control diet + 0.1% of a *Lactobacillus*-based probiotic; ^2^ LPr-0.2% = Control diet + 0.2% of a *Lactobacillus*-based probiotic; ^3^ BPo-0.2% = Control diet + 0.2% of a *Bifidobacterium*-based postbiotic.

**Table 2 microorganisms-12-01786-t002:** Average daily gain (ADG, g/d) of pigs from weaning to market weight in four dietary treatment groups: Control diet, Control diet + 0.1% of a *Lactobacillus*-based probiotic, Control diet + 0.2% of a *Lactobacillus*-based probiotic, and Control diet + 0.2% of a *Bifidobacterium*-based postbiotic.

Time Period	Control	LPr-0.1% ^1^	LPr-0.2% ^2^	BPo-0.2% ^3^	SEM	*p*-Value
D0–D10	147.53	142.70	134.63	133.78	6.13	0.84
D10–D21	326.95	311.30	327.24	322.74	6.10	0.78
D21–D47	617.72	617.52	616.84	625.83	4.95	0.91
D47–D70	926.51	946.28	933.96	941.30	9.21	0.89
D70–D105	1020.86	1005.86	1015.61	1020.11	8.65	0.92
D105–D135	994.61	1006.03	989.60	979.40	10.50	0.84

^1^ LPr-0.1% = Control diet + 0.1% of a *Lactobacillus*-based probiotic; ^2^ LPr-0.2% = Control diet + 0.2% of a *Lactobacillus*-based probiotic; ^3^ BPo-0.2% = Control diet + 0.2% of a *Bifidobacterium*-based postbiotic.

**Table 3 microorganisms-12-01786-t003:** Average daily feed intake (ADFI, g/d) of pigs from weaning to market weight in four dietary treatment groups: Control diet, Control diet + 0.1% of a *Lactobacillus*-based probiotic, Control diet + 0.2% of a *Lactobacillus*-based probiotic, and Control diet + 0.2% of a *Bifidobacterium*-based postbiotic.

Time Period	Control	LPr-0.1% ^1^	LPr-0.2% ^2^	BPo-0.2% ^3^	SEM	*p*-Value
D0–D10	153.62 ^a^	176.26 ^b^	175.13 ^b^	171.64 ^b^	3.38	0.05
D10–D21	553.60	590.11	568.30	584.00	14.04	0.80
D21–D47	960.62	916.27	932.38	908.26	11.05	0.36
D47–D70	1684.63	1744.74	1645.82	1731.07	26.02	0.53
D70–D105	2365.07	2364.43	2380.40	2293.72	31.45	0.78
D105–D135	2403.50	2398.90	2394.34	2381.32	30.34	0.99

^1^ LPr-0.1% = Control diet + 0.1% of a *Lactobacillus*-based probiotic; ^2^ LPr-0.2% = Control diet + 0.2% of a *Lactobacillus*-based probiotic; ^3^ BPo-0.2% = Control diet + 0.2% of a *Bifidobacterium*-based postbiotic; ^a,b^ Different superscripts in the same row indicate that groups were significantly different.

**Table 4 microorganisms-12-01786-t004:** Mean gain-to-feed ratio (G/F) of pigs from weaning to market weight in four dietary treatment groups: Control diet, Control diet + 0.1% of a *Lactobacillus*-based probiotic, Control diet + 0.2% of a *Lactobacillus*-based probiotic, and Control diet + 0.2% of a *Bifidobacterium*-based postbiotic.

Time Period	Control	LPr-0.1% ^1^	LPr-0.2% ^2^	BPo-0.2% ^3^	SEM	*p*-Value
D0–D10	1.14	1.32	1.35	1.35	0.11	0.51
D10–D21	1.72	1.94	1.76	1.85	0.13	0.69
D21–D47	1.56	1.50	1.51	1.45	0.03	0.28
D47–D70	1.83	1.85	1.76	1.84	0.06	0.76
D70–D105	2.32	2.35	2.35	2.25	0.06	0.63
D105–D135	2.42	2.39	2.43	2.44	0.07	0.97

^1^ LPr-0.1% = Control diet + 0.1% of a *Lactobacillus*-based probiotic. ^2^ LPr-0.2% = Control diet + 0.2% of a *Lactobacillus*-based probiotic. ^3^ BPo-0.2% = Control diet + 0.2% of a *Bifidobacterium*-based postbiotic.

**Table 5 microorganisms-12-01786-t005:** Histological analysis of jejunal and ileal tissues collected on D10 from pigs on four dietary treatment groups that were fed Control diet, Control diet + 0.1% of a *Lactobacillus*-based probiotic, Control diet + 0.2% of a *Lactobacillus*-based probiotic, and Control diet + 0.2% of a *Bifidobacterium*-based postbiotic.

	Control	LPr-0.1% ^1^	LPr-0.2% ^2^	BPo-0.2% ^3^	SEM	*p*-Value
Jejunum						
Villus height (mm)	342	350	386	349	18	0.36
Crypt depth (mm)	314	303	291	286	11	0.30
VH/CD ^4^	1.11	1.17	1.33	1.22	0.06	0.15
Ileum						
Villus height (mm)	255	289	289	296	17	0.33
Crypt depth (mm)	264	274	241	253	14	0.41
VH/CD ^4^	0.99 ^a^	1.04 ^b^	1.21 ^c^	1.18 ^c^	0.05	0.02

^1^ LPr-0.1% = Control diet + 0.1% of a *Lactobacillus*-based probiotic; ^2^ LPr-0.2% = Control diet + 0.2% of a *Lactobacillus*-based probiotic; ^3^ BPo-0.2% = Control diet + 0.2% of a *Bifidobacterium*-based postbiotic; ^4^ Villus height/crypt depth ratio; ^a,b,c^ Different superscripts in the same row indicate that groups were significantly different.

**Table 6 microorganisms-12-01786-t006:** Mean relative abundance (%) of the main bacterial phyla and families identified at two different time points (D10 and D47) in pigs in three dietary treatment groups: Control diet, Control diet + 0.2% of a *Lactobacillus*-based probiotic, and Control diet + 0.2% of a *Bifidobacterium*-based postbiotic.

Taxa	CO-D10 ^1^	LPr-D10 ^2^	BPo-D10 ^3^	CO-D47 ^4^	LPr-D47 ^5^	BPo-D47 ^6^
Bacillota ^$^	92.45	89.33	91.79	89.15	90.84	89.69
Lachnospiraceae	57.13 ^a^	50.10 ^a^	48.30 ^a^	12.67 ^b^	13.97 ^b^	12.58 ^b^
Lactobacillaceae	3.53 ^a^	10.36 ^ab^	10.78 ^b^	25.79 ^c^	33.63 ^d^	36.17 ^cd^
Streptococcaceae	0.15 ^a^	0.10 ^a^	0.31 ^a^	26.36 ^b^	20.69 ^bc^	14.85 ^c^
Clostridiaceae 1 ^%^	5.72	4.42	3.86	8.61	8.35	9.89
Erysipelotrichaceae	6.21 ^ac^	8.18 ^a^	10.05 ^a^	2.17 ^b^	3.77 ^bc^	3.52 ^bc^
C. Incertae Sedis XIII	4.36 ^a^	3.10 ^a^	2.50 ^a^	0.40 ^b^	0.88 ^c^	0.74 ^bc^
Ruminococcaceae	9.79 ^a^	8.49 ^ab^	9.13 ^ab^	6.48 ^ab^	5.12 ^b^	6.11 ^ab^
Other Firmicutes ^#^	5.57	4.57	6.86	6.68	4.43	5.84
Bacteroidota	1.69 ^a^	3.95 ^ab^	3.57 ^ab^	7.47 ^b^	4.87 ^b^	6.35 ^b^
Prevotellaceae	1.12 ^a^	3.36 ^a^	1.51 ^a^	7.15 ^b^	4.41 ^b^	6.03 ^b^
Other Bacteroidota ^#^	0.57	0.59	2.06	0.33	0.46	0.32
Actinomycetota ^%^	0.97	1.17	1.45	1.40	1.82	0.53
Other Phyla ^*#^	0.58	0.18	0.54	0.71	0.44	0.98
Unclassified Bacteria ^#^	4.30	5.37	2.64	1.27	2.03	2.46

^1^ CO-D10 = Samples from pigs fed a Control diet on D10; ^2^ LPr-D10 = Samples from pigs fed a Control diet + 0.2% of a *Lactobacillus*-based probiotic on D10; ^3^ BPo-D10 = Samples from pigs fed a Control diet + 0.2% of a *Bifidobacterium*-based postbiotic on D10; ^4^ CO-D47 = Samples from pigs fed a Control diet on D47; ^5^ LPr-D47 = Samples from pigs fed a Control diet + 0.2% of a *Lactobacillus*-based probiotic on D47; ^6^ BPo-D47 = Samples from pigs fed a Control diet + 0.2% of a *Bifidobacterium*-based postbiotic on D47; ^a,b,c,d^ Different superscripts in the same row indicate that groups were significantly different. ^%^ Differences between groups could not be resolved using the Wilcoxon pairwise test. ^$^ No difference using the Kruskal–Wallis test. ^#^ Statistical testing not performed due to group heterogeneity. * Combined abundance of other valid phyla found at an average representation lower than 1% across all groups: Campylobacterota, Candidatus Saccharimonadota, Fibrobacterota, Mycoplasmatota, Planctomycetota, Pseudomonadota, Spirochaetota, Synergistota, Verrucomicrobiota. C. Incertae Sedis XIII: Clostridiales Incertae Sedis XIII.

**Table 7 microorganisms-12-01786-t007:** Mean of alpha diversity indices at two different time points (D10 and D47) in pigs in three dietary treatment groups: Control diet, Control diet + 0.2% of a *Lactobacillus*-based probiotic, and Control diet + 0.2% of a *Bifidobacterium*-based postbiotic. No statistical differences were found (*p* > 0.05).

Diversity Index	CO-D10 ^1^	LPr-D10 ^2^	BPo-D10 ^3^	CO-D47 ^4^	LPr-D47 ^5^	BPo-D47 ^6^
Observed OTUs	903	688	811	888	768	905
Chao	3204	2353	2570	3092	2413	3119
Ace	6460	4495	4987	6174	4632	6227
Shannon	3.70	3.43	3.73	3.67	3.50	3.59
Simpson	0.16	0.14	0.13	0.13	0.14	0.15

^1^ CO-D10 = Samples from pigs fed a Control diet on D10. ^2^ LPr-D10 = Samples from pigs fed a Control diet + 0.2% of a *Lactobacillus*-based probiotic on D10. ^3^ BPo-D10 = Samples from pigs fed a Control diet + 0.2% of a *Bifidobacterium*-based postbiotic on D10. ^4^ CO-D47 = Samples from pigs fed a Control diet on D47. ^5^ LPr-D47 = Samples from pigs fed a Control diet + 0.2% of a *Lactobacillus*-based probiotic on D47. ^6^ BPo-D47 = Samples from pigs fed a Control diet + 0.2% of a *Bifidobacterium*-based postbiotic on D47.

**Table 8 microorganisms-12-01786-t008:** Mean relative abundance (%) of abundant OTUs identified at two different time points (D10 and D47) in pigs in three dietary treatment groups: Control diet, Control diet + 0.2% of a Lactobacillus-based probiotic, and Control + 0.2% of a Bifidobacterium-based postbiotic. The respective abundances of OTUs shown were found to vary across treatment groups (*p* < 0.05).

OTUs	CO-D10 ^1^	LPr-D10 ^2^	BPo-D10 ^3^	CO-D47 ^4^	LPr-D47 ^5^	BPo-D47 ^6^
Ssd-00001	0.64 ^a^	1.96 ^a^	5.10 ^a^	17.70 ^b^	25.45 ^b^	29.58 ^b^
Ssd-00003	0.35 ^a^	2.14 ^a^	0.64 ^a^	3.68 ^b^	1.78 ^b^	2.34 ^b^
Ssd-00011	<0.01 ^a^	<0.01 ^a^	0.06 ^a^	1.06 ^b^	0.70 ^b^	0.31 ^b^
Ssd-00013	0.67 ^a^	0.03 ^ab^	1.33 ^a^	<0.01 ^ab^	<0.01 ^b^	<0.01 ^ab^
Ssd-00014	0.10 ^a^	0.04 ^a^	0.03 ^a^	1.41 ^b^	1.07 ^b^	1.49 ^b^
Ssd-00019	1.29	3.81	0.89	4.16	4.60	3.00
Ssd-00039	0.07 ^a^	0.07 ^a^	0.07 ^a^	23.03 ^b^	18.66 ^bc^	12.70 ^c^
Ssd-00064	32.11 ^a^	30.03 ^a^	26.30 ^a^	1.89 ^b^	1.71 ^b^	1.64 ^b^
Ssd-00123	0.16 ^ab^	0.37 ^ab^	1.04 ^a^	0.07 ^b^	0.32 ^ab^	0.24 ^ab^
Ssd-00134	2.10 ^a^	1.53 ^a^	1.37 ^a^	7.97 ^b^	7.46 ^b^	8.86 ^b^
Ssd-00308	2.93 ^a^	4.73 ^a^	4.44 ^a^	1.01 ^b^	0.68 ^b^	0.96 ^b^
Ssd-00314	1.17	0.42	1.13	0.31	0.21	0.30
Ssd-00577	3.52 ^a^	3.04 ^a^	2.34 ^ab^	0.95 ^b^	1.23 ^b^	1.08 ^b^
Ssd-00950	2.81 ^a^	2.05 ^a^	1.17 ^abc^	0.23 ^b^	0.61 ^c^	0.44 ^bc^
Ssd-01077	1.19 ^a^	4.25 ^ab^	0.33 ^ab^	0.02 ^ab^	0.03 ^ab^	0.02 ^b^
Ssd-01079	0.17 ^ab^	0.01 ^b^	1.18 ^ab^	0.04 ^ab^	0.11 ^a^	0.30 ^a^
Ssd-01187	0.32 ^a^	0.13 ^a^	0.06 ^a^	0.13 ^a^	1.73 ^b^	1.35 ^b^
Ssd-01245	0.40 ^ab^	1.23 ^ab^	0.86 ^a^	0.01 ^b^	<0.01 ^b^	<0.01 ^b^

^1^ CO-D10 = Samples from pigs fed a Control diet on D10. ^2^ LPr-D10 = Samples from pigs fed a Control diet + 0.2% of a *Lactobacillus*-based probiotic on D10. ^3^ BPo-D10 = Samples from pigs fed a Control diet + 0.2% of a *Bifidobacterium*-based postbiotic on D10. ^4^ CO-D47 = Samples from pigs fed a Control diet on D47. ^5^ LPr-D47 = Samples from pigs fed a Control diet + 0.2% of a *Lactobacillus*-based probiotic on D47. ^6^ BPo-D47 = Samples from pigs fed a Control diet + 0.2% of a *Bifidobacterium*-based postbiotic on D47. ^a,b,c^ Different superscripts in the same row indicate that groups were significantly different based on the Wilcoxon pairwise test. Taxonomic affiliations of OTUs shown in the table can be found in Appendix A.

**Table 9 microorganisms-12-01786-t009:** Mean relative abundance (%) of abundant OTUs identified at two different time points (D10 and D47) in pigs in three dietary treatment groups: Control diet, Control diet + 0.2% of a Lactobacillus-based probiotic, and Control + 0.2% of a Bifidobacterium-based postbiotic. The respective abundances of OTUs shown did not vary across treatment groups (*p* > 0.05).

OTUs	CO-D10 ^1^	LPr-D10 ^2^	BPo-D10 ^3^	CO-D47 ^4^	LPr-D47 ^5^	BPo-D47 ^6^
Ssd-00002	0.06	1.65	1.62	0.21	0.14	0.04
Ssd-00117	1.30	1.45	0.86	0.50	0.72	0.76
Ssd-00125	0.29	0.38	0.72	1.61	0.47	0.82
Ssd-00165	1.51	1.71	1.09	1.45	1.47	1.15
Ssd-00277	1.23	0.53	0.34	0.03	0.12	0.10
Ssd-00318	1.45	1.92	1.93	1.09	0.85	0.47
Ssd-00409	1.33	0.51	0.27	0.18	0.25	0.25
Ssd-00450	1.02	1.10	1.03	0.51	0.92	0.66
Ssd-00495	1.37	0.72	1.90	0.37	0.33	0.31
Ssd-00815	0.08	0.08	1.21	0.02	0.02	0.01
Ssd-00892	0.61	1.20	0.39	0.32	0.17	0.17
Ssd-01061	0.34	0.05	1.53	0.04	0.19	0.03
Ssd-01102	0.39	0.30	1.27	0.14	0.15	0.08
Ssd-01122	0.44	1.29	0.15	0.05	0.07	0.05
Ssd-01134	0.96	1.39	0.44	0.41	0.33	0.22
Ssd-01243	0.80	2.03	0.34	0.01	0.03	0.06

^1^ CO-D10 = Samples from pigs fed a Control diet on D10. ^2^ LPr-D10 = Samples from pigs fed a Control diet + 0.2% of a *Lactobacillus*-based probiotic on D10. ^3^ BPo-D10 = Samples from pigs fed a Control diet + 0.2% of a *Bifidobacterium*-based postbiotic on D10. ^4^ CO-D47 = Samples from pigs fed a Control diet on D47. ^5^ LPr-D47 = Samples from pigs fed a Control diet + 0.2% of a *Lactobacillus*-based probiotic on D47. ^6^ BPo-D47 = Samples from pigs fed a Control diet + 0.2% of a *Bifidobacterium*-based postbiotic on D47. Taxonomic affiliations of OTUs shown in the table can be found in Appendix A.

## Data Availability

Raw sequence data are available from the NCBI Sequence Read Archive under BioProject PRJNA1128692.

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
