# Peer review of "Impact of Lactobacillus- and Bifidobacterium-Based Direct-Fed Microbials on the Performance, Intestinal Morphology, and Fecal Bacterial Populations of Nursery Pigs"

_microorganisms, 2024, doi:10.3390/microorganisms12091786_

Round 1

Reviewer 1 Report

Comments and Suggestions for Authors

The paper investigates the effects of direct-fed microbials (DFMs) on the performance, gut morphology, and bacterial composition of weaned pigs. It reports that DFM supplementation improves intestinal morphology and impacts gut microbial communities, identifying specific bacterial strains affected by the treatment. The findings contribute to understanding how DFMs influence swine health and performance.

Overall, the paper is well-structured and provides valuable data, particularly by identifying specific bacterial OTUs influenced by DFM supplementation. However, there are areas where the analysis could be more in-depth. For example, the discussion on performance metrics would benefit from a clearer explanation of why certain results were observed, especially when they contradict previous studies. Additionally, some terminology and presentation issues need addressing for clarity.

Other minor issues include:

  • Line 2: Add a hyphen between “direct” and “fed” to read "direct-fed."

  • Lines 13-14: The term "subclinical dosage" is vague; please rephrase to use a more precise term.

  • Line 27: The term "proxy" is vague; consider rephrasing to use a more precise term.

  • Line 27: Replace "At the family level" with "At the bacterial family level."

  • Lines 31-32: Clarify what is meant by "uncharacterized strains of Congobacterium massiliense and Absicoccus porci"; specify if these are "putative strains" or "strains with limited characterization."

  • Lines 33-34: The phrase “opposite composition pattern” is unclear; please provide a more precise explanation.

  • Line 58: Clarify the meaning of “subtherapeutic doses” by specifying the dose range.

  • Line 75: The statement "enhance gut health and barrier function by providing energy to epithelial cells" is vague; provide a clearer scientific explanation.

  • Line 107: The phrase "until DFM stocks were exhausted" is unscientific and lacks precision; replace it with a specific time frame.

  • Line 123: Correct the parentheses to read: (current pen weight - previous pen weight + removed pig weights).

  • Lines 137-138: Provide a clearer description of the euthanasia method, including what a “non-penetrating captive bolt gun” is and ensure it meets ethical and scientific standards. Include the regulations followed.

  • Lines 216-218, 239, 262-263, 354-355: Correct formatting errors with table captions and sub-notes.

Reviewer 2 Report

Comments and Suggestions for Authors

This study investigated the effects of Lactobacillus- and Bifidobacterium-based direct fed microbials on the performance, intestinal morphology, and fecal bacterial populations of nursery pigs.  The experimental design is reasonable, the experimental data is detailed, and the results are credible.

However, in Materials and Methods, two kinds of probiotics used in this study need to be introduced in detail, including their composition content, processing or coating methods. Because these would affect the experimental results.

In addition, the author needs to analyze and discuss some key experimental results in the Discussion. For example, how about the economic benefit of these probiotic productions? Can these probiotics safely pass through stomach acid? What is the relationship between the change of intestinal flora and the addition of these probiotics?

Line 103-106: Please state the main ingredients and content of these probiotic productions, e.g. the content of Lactobacillus, how many CFU/g?

Line 135: Here the pig was killed and an intestinal sample was taken, why was a sample of the intestinal contents not taken for microbiological?

Table 6: Row 3: For the relative abundance of Lactobacillaceae, why was the addition of the Bifidobacterium-based postbiotic group higher than the addition of the Lactobacillus-based postbiotic group? Although the difference is not significant.

Line 353-356: Please note the location of the Table. Notes should be below the Table.

Line 578-579: “The results presented in this report revealed that supplementation with individual DFMs could benefit performance and gut health in nursery pigs” This conclusion needs to be rewritten. For some key growth traits, none of them are significantly different. ADFI only improved from 0 to 10, but eating more did not bring more weight gain. Most of the results of this study were insignificant differences between treatments. Focusing on the few results that were different and ignoring the majority that were not significantly different would introduce some bias.

The highlight of this study lies in the authenticity of its experimental results and the fact that the experimental design has two time points to test, making the results more credible. These negative experimental results also need to be reported, otherwise all effective studies will be published, which will mislead readers.
